# A Nationwide Survey of the Attitudes toward the Use of Dietary Supplements among Japanese High-School Students

**DOI:** 10.3390/nu11071469

**Published:** 2019-06-28

**Authors:** Chiharu Nishijima, Etsuko Kobayashi, Yoko Sato, Tsuyoshi Chiba

**Affiliations:** Department of Food Function and Labeling, National Institute of Health and Nutrition, National Institutes of Biomedical Innovation, Health and Nutrition, 1-23-1 Toyama, Shinjuku-ku, Tokyo 162-8636, Japan

**Keywords:** dietary supplement, high-school students, internet survey

## Abstract

We previously studied the prevalence of dietary supplement use in Japanese high-school students by conducting a study on mothers. However, there is often a discrepancy between mothers’ understanding and children’s attitudes. Thus, we conducted an internet survey of high-school students to clarify their attitudes toward the use of dietary supplements. An invitation to the internet survey was e-mailed to registrants of a research company aged 15 to 18 years. A total of 1031 students (276 male, 755 female) answered the questionnaire on a first come, first served basis. The participants were classified according to the purchasers of their supplements: students who purchased supplements themselves were defined as active users, and others were defined as passive users. The prevalence of dietary supplement use was 30.8% in males and 26.7% in females. Among the users, 42.4% of males and 43.8% of females were active users. Males used supplements for health regardless of active or passive use. However, in females, more active users (40.9%) used supplements for weight loss than passive users (20.4%); the corresponding prevalence was 2.3% in our previous report on mothers’ understanding of their daughters. The most frequently used source of information for active users was the internet, whereas for passive users it was family. The frequency of adverse events was 9.4% in males and 14.4% in females, with gastrointestinal symptoms being the most commonly experienced events. Our results suggest that dietary education, including healthy eating and the appropriate use of dietary supplements, should be provided to high-school students.

## 1. Introduction

The use of dietary supplements has been increasing in Japan, with the prevalence estimated to be approximately 60% in healthy adults [1], 55%–70% in adult patients [1], and 32% in college students [2]. The use of such supplements has expanded to preschoolers and school-aged children, who are still in a state of physical and behavioral development and need to acquire appropriate dietary habits [3,4]. Previously, in our nationwide survey on mothers of children, we reported that supplement use among school-aged children varied from 15.3% in 1st–3rd grade in primary school to 29.2% in high school, with the proportion increasing with age [4]. Supplement use in children has been shown to be associated with older age of mothers and higher levels of education or eating awareness [3,5,6]. Our previous study indicated that supplement use was influenced by mothers’ perceptions of dietary supplements that overestimated their safety. Therefore, providing education on dietary supplements to mothers could prevent the inappropriate use of supplements and in turn prevent adverse events among children.

In high-school years, children typically become busy with school, sports, spending time with friends, and preparing for college entrance exams; hence, they communicate less with their mothers. Some studies have determined the global prevalence of dietary supplement use in adolescent populations [6,7,8,9,10]. It has also been observed that some children in this age group engage in unfavorable behaviors, such as disordered eating, which is related to supplement use [11,12]. Furthermore, they spend more time outside of the home, and are thus more likely to be influenced by others. This influence is expected to enrich their thoughts and establish their own perspectives; however, it may also lead them to misunderstandings or improper behavior.

In Japan, the monthly cost of a commercially available dietary supplement varies from around 300 yen to several thousand yen (equivalent to around 3 USD to several tens of USD), which high-school students can afford with the help of their allowance or salary from part-time jobs. They may purchase supplements on their own without telling their mothers. Therefore, in this study, we conducted a nationwide survey of high-school students to examine the prevalence of active supplement users, i.e., those who purchased supplements on their own, and compared their perceptions and usages of dietary supplements with those of passive supplement users, i.e., those who were supplied dietary supplements by others, and with mothers’ reports obtained in our previous study.

## 2. Materials and Methods 

### 2.1. Online Survey Procedure

The present study was conducted with the approval of the Research Ethics Committee of the National Institutes of Biomedical Innovation, Health and Nutrition (No. 119; approval date: 15 May 2017), in accordance with the Declaration of Helsinki. An internet survey was conducted by Cross Marketing Inc. (Tokyo, Japan), a survey research company, from 20 to 26 December 2017. The company holds over 4 million Japanese registrants with annually confirmed sociodemographic information, and eliminates fraudulent respondents when identified.

We submitted a questionnaire with a request of at least 1000 responses. The research company sent an invitation e-mail to participate in the survey to 46,019 of its registrants (17,583 males, 28,436 females) comprising high-school students aged 15 to 18 years; the number of invitations was decided by the research company based on their predicted response rate. The survey was administered in the following order: the questions started with general participant questions; then, further questions were asked only to dietary supplement users. The responses were collected and their quality was checked by the research company on a first come, first served basis until the number of complete responses reached 1000. A total of 1031 responses (276 males, 755 females) from regions all over Japan were considered eligible.

### 2.2. Questionnaires

The questionnaire was designed based on one that was previously administered to mothers of school-aged children, in order to assess their behaviors toward supplement use [4]. A dietary supplement was defined as a product that was considered to have beneficial effects on human health (e.g., vitamins, minerals, fish oil, and amino acids), excluding conventional foods (e.g., vegetables, fruits, and milk). 

The general participant questions covered dietary supplement use and perceptions regarding dietary supplements. Respondents who answered “currently using” (*n* = 127) and “previously used” (*n* = 159) were considered to be “supplement users”, and those who answered “never used” (*n* = 745) were considered to be “supplement non-users”. These 14 questions included a five-point scale ranging from “strongly agree” to “strongly disagree”; “strongly disagree” was deemed to be the most preferred answer [4]. The most preferred answer was determined considering the current dietary situation in Japan; for example, it is highly recommended to consume adequate amounts of nutrients from regular meals without using health foods, including dietary supplements. Additionally, dietary supplements and drugs are strictly separated by Japanese regulation; to treat diseases caused by nutrient deficiency, vitamin and mineral drugs are used rather than supplements. The general participant questions also included demographic information, such as school grade, area of residence, and sports or exercise habits (yes/no). Additionally, supplement users were asked further questions regarding the purchasers of the supplements—those who answered “oneself” were defined as active users, while those who responded “parents”, “grandparents”, or “distributed in the school program” were defined as passive users—as well as the purpose of use, the sources of information on supplement products, the number of products concomitantly used, the types of dietary supplements, the place of purchase, the awareness of the effectiveness of supplements, adverse event experiences, and responses to adverse events. 

### 2.3. Statistical Analysis

Differences in the demographic information and the perception questions among active users, passive users, and non-users were examined using the chi-square test. The question items for which significant differences were observed were further tested by performing residual analysis with the Holm adjustment. The responses of active and passive supplement users were compared using either the chi-square test or Fisher’s exact test. Statistical analyses were performed using R version 3.4.3, and p-values less than 0.05 were considered to indicate significance.

## 3. Results

### 3.1. Characteristics

Table 1 shows the characteristics of participants (*n* = 1031; 276 males, 755 females) according to sex and the use of dietary supplements. The prevalence of supplement use was 30.8% (13.0% were active users and 17.8% were passive users) in males and 26.6% (11.7% active users; 15.0% passive users) in females. Among the supplement users, active users comprised 42.4% in males (36/85) and 43.8% in females (88/201). Among females, supplement use was more common in the 3rd grade (32.5%) than in the 1st (24.0%) or 2nd (22.6%) grade (*p* = 0.01). Among males, supplement use was more common among those reported to be involved in sports or exercise habits (38.9%) than in those reported not to be involved in sports or exercise (24.4%) (*p* = 0.03). The logistic regression also showed a significant association between dietary supplement use and academic grades in females (*p* < 0.01) and between dietary supplement use and sports or exercise habits in males (*p* = 0.03). 

### 3.2. Perceptions about Dietary Supplements

The respondents’ perceptions about dietary supplements according to sex and the use of supplements are shown in Table 2. By setting the answer “strongly disagree” as the most preferred, the trend indicated that non-users mostly answered “strongly disagree”, followed by passive users. More active users agreed or strongly agreed on a majority of the topics—11 and 13 out of the 14 topics in males and females, respectively. Nearly half of active users agreed (strongly agree or agree) with the following statements regarding the safety of dietary supplements: “Dietary supplements are safe because they are just foods” (47.2% of males, 43.2% of females); “Dietary supplements made from natural ingredients or herbs are safe” (47.2%, 53.5%); “Dietary supplements made from foods are safe” (38.9%, 51.1%). Additionally, nearly half of active users stated (agreed or strongly agreed) that they wanted to use dietary supplements that were highly recommended by others (47.2%, 53.4%). Over half of the female active users agreed with the statements, “I want to use weight-loss or muscle-building dietary supplements” (54.5%) and “Children should learn about dietary supplements at school” (56.8%). In males, more active users agreed with the statement that, “Dietary supplements can help prevent diseases” (*p* < 0.01), whereas female active users agreed with the statements that “Dietary supplements can be used concomitantly with drugs” (*p* < 0.01) and “Pregnant women should take dietary supplements to supplement nutrition” (*p* < 0.01).

### 3.3. Information Sources and Ways of Obtaining Dietary Supplements

Active users gained information mostly from the internet—55.6% of males and 62.5% of females—while most passive users stated “Family” as their source of information (Table 3). For all respondents, the second most popular source of information was television in both males and females. Male active users mostly gained information from “Product labels” (30.6%, *p* = 0.02) and “Friends or acquaintances” (25.0%, *p* = 0.03), while female active users relied more on the advice of “Pharmacists or drug store clerks” (25.0%, *p* < 0.01) and “Point-of-purchase displays” (23.9%, *p* = 0.01) than female passive users.

For active users, the most common way to purchase dietary supplements, for both males and females, was “Pharmacy or drugstores” (75.0% and 65.9%, respectively), followed by the internet (44.4% and 28.4%, respectively). For males, the third most common way was “Supermarket” (25.0%), while for females it was “Mail order (except internet shopping)” (13.6%).

### 3.4. The Purpose of Dietary Supplement Use and the Types of Supplements Used

The purpose of dietary supplement use and the types of supplements used are shown in Table 4 and Table 5, respectively. In males, more active users used supplements for the purpose of “Maintenance of health” (*p* = 0.01), to “Enhance stamina” (*p* = 0.03), and to “Enhance athletic performance” (*p* < 0.01), with the proportions being two to three times larger than those of male passive users. On the other hand, in females, twice as many active users used supplements for the purpose of “Weight loss” than did passive users (*p* < 0.01). 

Regarding the types of supplements used, i.e., between vitamins and minerals, individual minerals were used more by active users compared to passive users of the same sex, and were used more by males (*p* = 0.01) than by females (*p* = 0.03). The use of “Protein/Amino acid” was reported by half of the male active users. In females, 20.5% of active users reported the use of “Weight loss” supplements and 15.9% reported the use of “Beauty” supplements, while in female passive users, only 6.2% and 0.9% reported the use of the respective supplements (both, *p* < 0.01).

### 3.5. Concomitant Use of Dietary Supplements and Awareness of Their Effectiveness

The majority of passive users were using only one product (59.2% of males, 64.6% of females), whereas active users were using two or more products concomitantly (61.1% of males, 50.0% of females; Table 6). In males, more active users (61.1%) felt that the products were effective and less felt null effects (2.8%) or did not know/were not sure about the effects (36.1%). On the other hand, in females, both active users and passive users felt similar levels of effectiveness (38.6% and 36.3%, respectively), while more active users felt null effects (20.5%) of the products compared to passive users (8.0%).

### 3.6. Experiences of Adverse Events and Responses to the Symptoms

The proportion of adverse events experienced with supplement use and the response actions to the adverse events are shown in Table 7. The prevalence of adverse events amounted to 9.4% in males and 14.4% in females. Gastrointestinal symptoms such as “diarrhea” and “nausea and vomiting” were experienced the most. Some females, but no males, reported “allergy-like symptoms” (10.3%), “fatigue” (6.9%), and “palpitations” (3.4%). Over one-third of males and almost half of females took no action to alleviate the symptoms. One quarter of males and females stopped using the supplements immediately. A total of 25.0% of males and 17.2% of females stated that they “Consulted a family member” about adverse events.

## 4. Discussion

In the present study, we investigated the use of dietary supplements among Japanese high-school students and compared the findings to a previous study involving the mothers of high-school students [4]. Since children of this age start to develop more independence, they may have their own thoughts regarding dietary supplements and may use supplements outside of parental control. The prevalence of supplement users in this survey (30.8% of males, 26.6% of females) is similar to that reported in the previous study on mothers (32.8% of males, 25.5% of females) [4] and is also in line with surveys of high-school students in Korea (25.9% of students) [6], the US (25.7%) [10], and Italy (34.8%) [9]. However, in the present study, approximately 40% of students were active users who bought supplements on their own. Compared to passive users and non-users, more active users believed that dietary supplements were safe and that word of mouth from other users was trustworthy. In our previous report, over a half of the mothers who gave supplements to their school-aged children had the same perception about dietary supplements. This indicates the possibility that mothers’ perspectives on dietary supplements are transmitted to their children. On the other hand, unlike the mothers, supplement users—particularly active users—in this survey wrongly believed that dietary supplements help prevent diseases. Approximately 20% of active users thought that dietary supplements could be used concomitantly with drugs.

The present study found that female active users were more optimistic about the safety of supplement use than female passive users and non-users. Many of them thought that people who need to be careful about their nutritional intake should take supplements, although diet modifications should be focused on first. Additionally, approximately one-fifth of female active users used supplements for weight loss. A previous study found that the prevalence of dieting behaviors among Japanese females aged 15–19 was 41% to 68.6% [13]. A desire to lose weight quickly without making an effort to change dietary and exercise habits may drive females to want to use weight-loss supplements while completely believing in their safety and having no knowledge of their potential dangers. Marketed weight-loss supplements are often adulterated with undeclared illegal drug ingredients, and mild to severe adverse effects have been reported for such products [14,15]. Despite the efforts of the Japanese government to inhibit the marketing of such illegal products, supplement marketplaces outside Japan are still accessible to consumers, who use the internet as an information source. Thus, Japanese women suffer from adverse events due to the use of illegal supplements [16,17]. 

Our previous research on mothers found that only 2.3% of their daughters aged 15–18 used weight-loss supplements, which is one tenth of the proportion of high-school females who reported using such supplements in the present study. On the other hand, the present study found that female students who were passive supplement users gained information from their family, and their purpose for use was in line with the report on mothers. The results of the present study suggest that female students who are active users of dietary supplements make their own decisions without their mothers’ involvement.

In males, sports and exercise habits were significantly associated with the use of dietary supplements. For active users, the purpose of using supplements was the maintenance of health, the enhancement of stamina, and the enhancement of athletic performance. Therefore, they used proteins/amino acids and individual minerals such as zinc, calcium, and iron. Our results for active users are consistent with previous reports which directly surveyed high-school students in Italy [9] and Slovenia [7]. By contrast, in accordance with studies of students in Korea [6] and Poland [8], the present study found that dietary supplement use in passive users was not associated with sports and exercise habits. Rather, they focused more on the supplementation of nutrients and the maintenance of health. Meanwhile, according to our previous study on mothers, the use of individual minerals accounted for 17.2% of their sons aged 15–18, whereas the use in passive users was 2.0%. Since some supplements are manufactured to resemble conventional foods such as gummy candies, other candies, and chewable tablets, passive users may have been unaware that they were taking supplements that they were given by others.

Our previous study of mothers found that the prevalence of adverse events in response to supplements in people of high-school age was 2.1%, which was the lowest occurrence among all the ages of school-aged children and contradicted the idea that the prevalence of dietary supplement use increases with age [4]. However, in this study, the prevalence of adverse events was found to be five to seven times larger than that reported by mothers in our previous study. Since less than one quarter of the students who took part in the present study consulted with a family member after the onset of adverse events, they may not have reported a change in their physical condition, especially when it was small enough to be handled by themselves. The present study found that males primarily experienced gastrointestinal symptoms, which is consistent with our previous study of mothers, and these symptoms may not be particularly serious. However, females experienced other symptoms, which may be a sign of serious injury such as liver failure [18]. In such cases, the first step should be to stop using supplements to prevent serious injury.

There seems to be a trend towards overly trusting the safety of dietary supplements, obtaining information primarily from the internet and television, and purchasing supplements of choice by active users. These findings are supported by a report [19] that found an association between media exposure and the use of dietary supplements to enhance appearance or physique in adolescents. Observing idealized male and female beauty standards in the media and being exposed to incorrect information and misleading advertisements could be a stimulus for adolescents to engage in unhealthy practices, such as the easy use of dietary supplements. The trend in our study was more evident in females. It has been established that the use of supplements to change one’s body is associated with symptoms of eating disorders in adolescent females [11,12]. Although the present study found that only 3.5% to 8.3% of dietary supplement users consulted healthcare professionals, the use of supplements should be considered together with the evaluation of dietary habits and nutritional intake. Moreover, in adults, weight-loss supplements are popular among women while performance-related supplements are popular among men. These products are often seen to be associated with adverse events. According to mothers’ understanding of their children reported in our previous study, the use of these products among high-school students was rare. However, by asking high-school students directly, the present study revealed that the use of supplements by high-school students was similar to that observed in adults. These findings imply that dietary education, including the appropriate usage of and safety concerns regarding dietary supplements, needs to be provided to high-school students who start to move out of parental control.

This study has some limitations. First, the survey was conducted by a survey company through the internet. In Japan, the internet utilization rate by the 13–19 year age group is stable at over 98%, and high-school students living in various regions participated in this survey. However, the response rate was very low in this age group, and the participants may, therefore, not have been representative of Japanese high-school students. Indeed, the research company estimated that almost 40,000 samples were required to obtain 1000 responders. Moreover, we believe that the present study is the first to report dietary supplement use that is focused on the level of activity of the user. Second, the information on unintentional supplement use by participants may not have been included, especially for passive users who did not pay attention to what supplements they were given. It is also possible that the use of supplements that resemble conventional foods was not declared by the participants. Third, the reports about adverse events were all self-reported, and the severity and causal relationships of the symptoms were not medically determined. Additionally, information about adverse events from the small number of males who participated in this study was not representative of the general population of supplement users. However, in this study and a previous study, we conducted nationwide surveys of both mothers and children. Thus, the difference between the mothers’ understanding and children’s attitudes toward dietary supplements can be defined. Surveying children enabled us to determine the current circumstances on the use of certain products, and their adverse events. The findings provide basic insight to create more suitable education programs for mothers and children, respectively.

## 5. Conclusions

The present study revealed that the prevalence of dietary supplement use among Japanese high-school students was 30.8% in males and 26.6% in females. Approximately 40% of dietary supplement users were active users who purchased supplements on their own and selected the products based on information obtained from the internet and television. Overall, the use of supplements by passive users was in agreement with a previous report on mothers. Among the active users, males used supplements mainly for health and sports purposes while females used them for health and weight-loss purposes. Since supplement users, and particularly active users, are vulnerable to convincing words on the internet and television and overlook the safety of supplements, dietary education, including healthy eating and the appropriate use of dietary supplements, needs to be provided.

## Figures and Tables

**Table 1 nutrients-11-01469-t001:** Characteristics of survey participants according to sex and the use of dietary supplements.

	Males								Females							
	All *n*	Supplement Users				Supplement Non-Users		*P*-Value	All *n*	Supplement Users				Supplement Non-Users		*P*-Value
		Active Users		Passive Users						Active Users		Passive Users				
All, *n* (%)																
	276	36	(13.0)	49	(17.8)	191	(69.2)		755	88	(11.7)	113	(15.0)	554	(73.4)	
Grade, *n* (%)								0.18								0.01
1st	87	7	(8.0)	12	(13.8)	68	(78.2)		229	24	(10.5)	31	(13.5)	174	(76.0)	
2nd	87	13	(14.9)	20	(23.0)	54	(62.1)		252	18	(**7.1**)	39	(15.5)	195	(77.4)	
3rd	102	16	(15.7)	17	(16.7)	69	(67.6)		274	46	(**16.8**)	43	(15.7)	185	(**67.5**)	
Sports or exercise habits, *n* (%)								0.03								0.77
Yes	121	22	(18.2)	25	(20.7)	74	(61.2)		201	24	(11.9)	33	(16.4)	144	(71.6)	
No	155	14	(9.0)	24	(15.5)	117	(75.5)		554	64	(11.6)	80	(14.4)	410	(74.0)	

Note: Percentages are shown in row ratio. Statistical analyses were conducted using a chi-square test. Significantly different proportions (*p* < 0.05) as determined by residual analysis are shown in bold.

**Table 2 nutrients-11-01469-t002:** Perceptions of dietary supplements according to sex and the use of dietary supplements (% of respondents).

		Males						Females					
		Strongly Agree	Agree	Neither Agree nor Disagree	Disagree	Strongly Disagree	*p*-Value	Strongly Agree	Agree	Neither Agree nor Disagree	Disagree	Strongly Disagree	*p*-Value
Dietary supplements are safe because they are just foods.	Active users	8.3	38.9	30.6	16.7	5.6	0.37	**12.5**	30.7	30.7	18.2	8.0	<0.01
	Passive users	4.1	30.6	40.8	16.3	8.2		3.5	25.7	39.8	27.4	**3.5**	
	Non-users	7.9	22.0	34.0	23.6	12.6		4.3	**16.6**	36.8	28.3	**13.9**	
Dietary supplements made from natural ingredients or herbs are safe.	Active users	2.8	44.4	38.9	5.6	8.3	0.28	**20.5**	33.0	29.5	11.4	5.7	<0.01
	Passive users	4.1	38.8	32.7	18.4	6.1		8.0	38.1	38.1	11.5	4.4	
	Non-users	7.3	26.2	40.3	16.2	9.9		**6.3**	**25.3**	37.9	19.3	11.2	
Dietary supplements made from foods are safe.	Active users	5.6	**33.3**	44.4	**13.9**	2.8	0.01	10.2	**40.9**	29.5	13.6	5.7	<0.01
	Passive users	6.1	**32.7**	30.6	24.5	6.1		4.4	30.1	44.2	18.6	2.7	
	Non-users	4.7	**17.8**	49.2	17.3	11.0		4.3	**22.7**	44.4	18.2	10.3	
Food additives should be avoided.	Active users	0	30.6	50.0	16.7	2.8	0.18	13.6	35.2	25.0	20.5	5.7	0.10
	Passive users	8.2	30.6	28.6	18.4	14.3		12.4	31.0	37.2	14.2	5.3	
	Non-users	7.3	30.9	40.8	9.4	11.5		11.6	30.0	38.4	10.8	9.2	
The effectiveness of commercial dietary supplements are confirmed and reliable.	Active users	2.8	33.3	52.8	**2.8**	8.3	0.02	8.0	15.9	48.9	18.2	9.1	0.03
	Passive users	8.2	12.2	49.0	24.5	6.1		2.7	16.8	52.2	22.1	6.2	
	Non-users	2.6	15.7	44.5	25.1	12.0		3.4	13.0	42.2	26.4	15.0	
I want to use dietary supplements that are highly recommended by users.	Active users	8.3	**38.9**	38.9	8.3	5.6	<0.01	**15.9**	**37.5**	29.5	12.5	4.5	<0.01
	Passive users	12.2	22.4	32.7	24.5	8.2		3.5	33.6	41.6	14.2	7.1	
	Non-users	3.7	14.7	44.0	18.3	19.4		6.7	**20.8**	35.0	20.6	**17.0**	
Products recommended by health professionals are effective.	Active users	8.3	16.7	47.2	25.0	2.8	0.02	9.1	15.9	37.5	23.9	13.6	0.06
	Passive users	6.1	26.5	28.6	26.5	12.2		4.4	14.2	37.2	32.7	11.5	
	Non-users	2.1	11.0	44.5	24.1	18.3		3.8	18.1	39.7	20.4	18.1	
Dietary supplements can be used concomitantly with drugs.	Active users	2.8	22.2	47.2	22.2	5.6	0.15	**9.1**	18.2	26.1	30.7	15.9	<0.01
	Passive users	2.0	10.2	42.9	28.6	16.3		3.5	13.3	38.9	31.0	13.3	
	Non-users	4.2	8.4	38.7	25.7	23.0		2.7	**7.6**	34.5	33.8	21.5	
Dietary supplements can help prevent diseases.	Active users	0	**41.7**	41.7	13.9	2.8	<0.01	6.8	14.8	26.1	31.8	20.5	0.06
	Passive users	**10.2**	16.3	36.7	18.4	18.4		3.5	17.7	36.3	24.8	17.7	
	Non-users	1.6	**11.0**	39.3	25.1	23.0		3.2	8.8	35.6	30.1	22.2	
Dietary supplements can treat diseases.	Active users	5.6	16.7	38.9	30.6	8.3	0.40	5.7	8.0	27.3	33.0	26.1	0.57
	Passive users	4.1	10.2	38.8	20.4	26.5		1.8	9.7	32.7	30.1	25.7	
	Non-users	3.7	7.9	34.6	27.2	26.7		3.1	5.8	35.0	29.6	26.5	
Children who are picky eaters should take dietary supplements to supplement nutrition.	Active users	2.8	27.8	36.1	25.0	8.3	0.07	9.1	17.0	27.3	27.3	19.3	0.05
	Passive users	8.2	12.2	36.7	22.4	20.4		5.3	15.0	31.0	26.5	22.1	
	Non-users	3.1	10.5	35.6	24.1	26.7		3.1	9.7	32.5	27.3	27.4	
Pregnant women should take dietary supplements to supplement nutrition.	Active users	5.6	25.0	36.1	25.0	8.3	0.17	**3.4**	**27.3**	**29.5**	**19.3**	**20.5**	<0.01
	Passive users	2.0	20.4	40.8	22.4	14.3		8.8	15.0	38.1	22.1	15.9	
	Non-users	4.2	9.9	43.5	20.9	21.5		3.6	9.7	36.6	25.6	24.4	
I want to use weight-loss or muscle-building dietary supplements.	Active users	8.3	25.0	41.7	16.7	8.3	0.06	**19.3**	**35.2**	26.1	13.6	**5.7**	<0.01
	Passive users	4.1	28.6	34.7	16.3	16.3		11.5	32.7	32.7	17.7	**5.3**	
	Non-users	5.8	11.5	36.1	22.5	24.1		**6.1**	**19.1**	34.1	20.2	**20.4**	
Children should learn about dietary supplements at school.	Active users	11.1	27.8	38.9	16.7	5.6	0.02	**26.1**	30.7	29.5	9.1	4.5	<0.01
	Passive users	16.3	34.7	26.5	14.3	8.2		12.4	28.3	39.8	13.3	6.2	
	Non-users	8.9	**15.7**	40.3	15.2	19.9		**9.7**	20.4	36.5	15.3	**18.1**	

The total number of males was 276 (36 active users, 49 passive users, 191 non-users) and the total number of females was 755 (88 active users, 113 passive users, 554 non-users). Note: Statistical analyses were conducted to examine the differences between active users, passive users, and non-users using a chi-square test. Significantly different proportions (*p* < 0.05) as determined by residual analysis are shown in bold.

**Table 3 nutrients-11-01469-t003:** A comparison of the sources of information regarding dietary supplements in active and passive users (% of respondents).

	Males				Females			
	All	Active Users	Passive Users	*p*-Value	All	Active Users	Passive Users	*p*-Value
Number	85	36	49		201	88	113	
Internet	37.6	55.6	24.5	<0.01	40.3	62.5	23.0	<0.01
Television	36.5	41.7	32.7	0.39	30.8	28.4	32.7	0.51
Family	25.9	16.7	32.7	0.10	23.9	8.0	36.3	<0.01
Product labels	18.8	30.6	10.2	0.02	13.9	9.1	17.7	0.08
Pharmacists or drug store clerks	15.3	22.2	10.2	0.13	16.4	25.0	9.7	<0.01
Stores (point-of-purchase displays)	5.9	11.1	2.0	0.08	16.4	23.9	10.6	0.01
Friends or acquaintances	15.3	25.0	8.2	0.03	8.5	9.1	8.0	0.78
Newspapers, magazines, flyers	12.9	19.4	8.2	0.13	11.9	10.2	13.3	0.51
Clinic (physicians, pharmacists, dietitians)	8.2	8.3	8.2	0.98	4.5	5.7	3.5	0.47
Radio	5.9	11.1	2.0	0.08	3.5	3.4	3.5	0.96
School teachers	4.7	5.6	4.1	0.75	2.0	2.3	1.8	0.80
Coaches of clubs	4.7	8.3	2.0	0.18	1.5	1.1	1.8	0.71
Inquiry to the manufacturer	1.2	0	2.0	–	0.5	1.1	0	–
Others	3.5	2.8	4.1	0.75	0.5	0	0.9	–

Note: Multiple answers. Statistical analyses were conducted among groups of purchasers using a chi-square or Fisher’s exact test (*p* < 0.05).

**Table 4 nutrients-11-01469-t004:** A comparison of the purpose of the use of dietary supplements in active and passive users (% of respondents).

	Males				Females			
	All	Active Users	Passive Users	*p*-Value	All	Active Users	Passive Users	*p*-Value
Number	85	36	49		201	88	113	
Supplementation of nutrients	47.1	44.4	49.0	0.68	58.2	55.7	60.2	0.52
Maintenance of health	47.1	63.9	34.7	0.01	39.8	36.4	42.5	0.38
Improvements to health	17.6	22.2	14.3	0.34	21.4	20.5	22.1	0.77
Enhance growth	27.1	30.6	24.5	0.53	7.5	8.0	7.1	0.81
Enhance stamina	29.4	41.7	20.4	0.03	5.5	5.7	5.3	0.91
Enhance athletic performance	21.2	36.1	10.2	<0.01	4.5	3.4	5.3	0.52
Weight loss	1.2	2.8	0	–	29.4	40.9	20.4	<0.01
Prevention of diseases	5.9	13.9	0	–	5.5	5.7	5.3	0.91
Treatment of diseases	2.4	0	4.1	–	5.5	6.8	4.4	0.46
Improve academic performance	1.2	0	2.0	–	3.5	3.4	3.5	0.96
Others	1.2	0	2.0	–	5.5	8.0	3.5	0.17

Note: Multiple answers. Statistical analyses were conducted among groups of purchasers using a chi-square or Fisher’s exact test (*p* < 0.05).

**Table 5 nutrients-11-01469-t005:** A comparison of the types of supplements used by active and passive users (% of respondents).

	Males				Females			
	All	Active Users	Passive Users	*p*-Value	All	Active Users	Passive Users	*p*-Value
Number	85	36	49		201	88	113	
**Vitamins/Minerals**								
Multi-vitamins and minerals	1.2	2.8	0	–	0.5	1.1	0	–
Multi-vitamins	5.9	11.1	2.0	0.28	4.5	6.8	2.7	0.45
Individual vitamins	12.9	11.1	14.3	0.15	22.4	26.1	19.5	0.60
Multi-minerals	1.2	0	2.0	–	0.5	1.1	0	–
Individual minerals	14.1	30.6	2.0	0.01	15.9	13.6	17.7	0.03
Any type	35.3	55.6	20.4	0.18	43.8	48.9	39.8	0.14
**Non-Vitamins/Non-Minerals**								
Protein/Amino acid	30.6	50.0	16.3	0.13	5.5	8.0	3.5	0.53
Weight loss	0	0	0	–	12.4	20.5	6.2	<0.01
Beauty ^1^	1.2	2.8	0	–	7.5	15.9	0.9	<0.01
Growth-promoting	7.1	5.6	8.2	0.23	0	0	0	–
Eye care	3.5	0	6.1	–	2.5	0	4.4	–
*n*-3 PUFA	4.7	2.8	6.1	0.19	2.0	2.3	1.8	0.85
Others ^2^	23.5	25.0	22.4	0.23	19.4	20.5	18.6	0.24

Abbreviation: PUFA; poly unsaturated fatty acid. ^1^ Includes products for skin care and body modification, excluding weight-loss related products. ^2^ Includes probiotics, aojiru (Japanese green vegetable juice), gamma aminobutyric acid, and various herbal products such as garlic, turmeric, and noni. Note: Multiple answers. Statistical analyses were conducted among groups of purchasers using a chi-square or Fisher’s exact test (*p* < 0.05).

**Table 6 nutrients-11-01469-t006:** A comparison of concomitant use and awareness of the effectiveness of dietary supplements in active and passive users (% of respondents).

	Males				Females			
	All	Active Users	Passive Users	*p*-Value	All	Active Users	Passive Users	*p*-Value
Number	85	36	49		201	88	113	
Concomitant use								
1 product	50.6	38.9	59.2	0.09	58.2	50.0	64.6	0.03
2 products	29.4	33.3	26.5		27.4	33.0	23.0	
3 products	15.3	25.0	8.2		10.9	10.2	11.5	
4 or more products	4.8	2.8	6.1		3.5	6.8	0.9	
Awareness								
Effective	43.5	61.1	30.6	0.01	37.3	38.6	36.3	0.02
Null	11.8	2.8	18.4		13.4	20.5	8.0	
Don’t know/Not sure	44.7	36.1	51.0		49.3	40.9	55.8	

Note: Statistical analyses were conducted among groups of purchasers using a chi-square or Fisher’s exact test (*p* < 0.05).

**Table 7 nutrients-11-01469-t007:** The experience of adverse events due to dietary supplement use and the actions undertaken in response to the events (% of respondents).

	Males	Females
Number	85	201
Has experienced adverse events	9.4	14.4
Symptoms ^1^		
Diarrhea	50.0	24.1
Nausea and vomiting	37.5	41.4
Headache	12.5	17.2
Stomachache	0	17.2
Constipation	0	10.3
Allergy-like symptoms (eczema and itching)	0	10.3
Fatigue	0	6.9
Palpitations	0	3.4
Others	12.5	3.4
Response Actions ^1^		
Did nothing	37.5	48.3
Stopped using supplements immediately	25.0	27.6
Consulted a family member	25.0	17.2
Complained to the manufacturer or the retail store	25.0	10.3
Reported the incident to governmental agencies ^2^	25.0	6.9
Reported the incident to a public health center	0	0
Went to a hospital	0	3.4
Others	12.5	0

Multiple answers. ^1^ Eight males and 29 females reported one or more symptoms. ^2^ Governmental agencies include the Ministry of Health, Labor and Welfare, the Consumer Affairs Agency, the Government of Japan, the National Consumer Affairs Center of Japan, and other consumer centers.

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
