# Peer review of "A Nationwide Survey of the Attitudes toward the Use of Dietary Supplements among Japanese High-School Students"

_nutrients, 2019, doi:10.3390/nu11071469_

Reviewer 1 Report

Thank you for the opportunity to review this article entitled "The attitude toward active and passive use of dietary supplements by Japanese high school students”.

 1) Title

·         Suggest to revise the title as follow: “A national survey on the attitude towards active and passive use of dietary supplements amongst Japanese high school students”

2) Abstract

·         Suggest to remove the reference to previous report in the abstract and provide the background/justification of the present study to close the gap in research/literature

·         Provide brief information on the methodology used in this study besides the questionnaire

 3) Introduction

·         Besides stating the prevalence of dietary supplements users in Japan for various age groups, suggest to include the consequences of under or overconsumption of dietary supplements amongst the adolescents to strengthen the justification of the present study to be conducted

 4) Materials and Methods

·         Best to move the sentence, Page 2, Line 62 to 64 “The present study was conducted with the approval of the Research Ethics Committee….” to the starting of Section 2

·         Kindly briefly explain on the study design and calculation of sample size as 1000 subjects – which software was used or what type of sampling method was used

 5) Discussion

·         Suggest to revise the sentence, Page 9, Line 188 to 190 as follow “In the present study, we investigated the use of dietary supplements amongst Japanese high school students and compared the findings to a previous report conducted with mothers of high school students [4].”

·         Suggest to reduce the length for Paragraph 2 (Page 9, Line 205 to 214) and Paragraph 5 (Page 10, Line 236 to 247) as there is repetition with the results section

·         While it is interesting to compare the findings to other studies, it would be important to relate the current findings and their implications on practice/ research in Japan

 6) Strengths of the paper

·         This is missing in the current manuscript and it is highly recommended to include the strengths besides stating the limitations

 7) Grammar and spelling

·         Please ensure the manuscript is proof read and edited as there are a number of grammatical errors throughout especially on the tenses used when citing the findings from other studies and selection of words could be improved for the purpose of scientific writing

Author Response

Answer to Reviewer.1

Thank you for valuable comments and useful suggestion to improve our manuscript. We carefully read them and changed our manuscript by following each comment.

 Title

1. Suggest to revise the title as follow: “A national survey on the attitude towards active and passive use of dietary supplements amongst Japanese high school students”

 (Response) Thank you for suggestion. We revised the title as the reviewer suggested, but we used “nationwide survey” instead of “national survey” because “nationwide survey” is used in our published researches in Nutrients. And, we deleted “active and passive” according to another reviewer’s comment. Thus, the revised title is “A nationwide survey on the attitude towards use of dietary supplements amongst Japanese high school students”. (Line 2-3)

 Abstract

2. Suggest to remove the reference to previous report in the abstract and provide the background/
justification of the present study to close the gap in research/literature

 (Response) We revised the description as follows: “We previously studied the prevalence of dietary supplement use in Japanese high-school students by conducting on mothers. However, there is often a discrepancy between mothers’ understanding and children’s attitudes. (Line 11-13)

 3. Provide brief information on the methodology used in this study besides the questionnaire

 (Response) We revised the description to “An invitation to the internet survey was e-mailed to registrants of research company aged 15 to 18 years, then 1,031 (boys: 276, girls: 755) answered the questionnaire on a first-come basis.” (Line 14-16)

 Introduction

4. Besides stating the prevalence of dietary supplements users in Japan for various age groups, suggest to include the consequences of under or overconsumption of dietary supplements amongst the adolescents to strengthen the justification of the present study to be conducted

 (Response) Thank you for suggestion. It is important to find out the consequences of under or overconsumption of dietary supplements that reviewer pointed out. However, we could not find the studies about it. So, we think that more studies are needed in this area especially in this age.

 Materials and Methods

5. Best to move the sentence, Page 2, Line 62 to 64 “The present study was conducted with the approval of the Research Ethics Committee….” to the starting of Section 2

 (Response) Thank you for suggestion. We moved the sentence to the starting of Section 2. (Line 60-62)

 6. Kindly briefly explain on the study design and calculation of sample size as 1000 subjects – which software was used or what type of sampling method was used

 (Response) Regarding sample size, we planned to collect 600 dietary supplement users of high school students as the same number as our previous survey conducted to mothers of high school students. Assuming the prevalence of supplement users approximately 30% from the previous reports, we requested for quotation for 2000 complete responses from high school students to several research companies which met certain standards. However, a response rate in this age is very low compared to that in general. Indeed, 46,019 registrants were needed to collect 1000 responses in this survey. Therefore, 1000 subjects were the maximum number within the limits of our budget.

To add explanation on the study design, we revised the description to:
“We submitted the questionnaire with a request of at least 1,000 complete responses. The research company sent an invitation e-mail to participate in the survey to its 46,019 registrants (boys: 17,583, girls: 28,436) comprising high school students, aged 15 to 18 years; the number of invitations was decided by the research company. The survey based on our questionnaire was administered in the following order; the questions started from the general participant questions; then, the further questions were asked only to dietary supplement users” (Line 67-72)
to explain the study design.

 Discussion

7. Suggest to revise the sentence, Page 9, Line 188 to 190 as follow “In the present study, we investigated the use of dietary supplements amongst Japanese high school students and compared the findings to a previous report conducted with mothers of high school students [4].”

 (Response) Thank you. We revised accordingly. “In the present study, we investigated the use of dietary supplements amongst Japanese high school students and compared the findings to a previous report conducted with mothers of high school students [4].” (Line 205-207)

 8. Suggest to reduce the length for Paragraph 2 (Page 9, Line 205 to 214) and Paragraph 5 (Page 10, Line 236 to 247) as there is repetition with the results section

 (Response) We deleted the repetition with the results section and rearranged Paragraph 2 and 3. “Girls of active users were more optimistic about the safety of supplement use than passive users and non-users. Thus, many of them think that those who need care about nutritional intake should take supplements, although diet modifications should be focused on first. In addition, approximately one-fifth of them used supplements for weight loss. There is a 41% to 68.6% prevalence of dieting behaviors among Japanese girls aged 15-19 [2]. A willingness to quickly lose weight, without making an effort to change their dietary and exercise habits, may drive girls for a desire to use weight loss supplements, while having undoubted belief in safety and no knowledge of its potential dangers. Marketed weight loss supplements are often adulterated with undeclared illegal drug ingredients, and mild to severe adverse effects of the product have been reported [3,4]. Despite the efforts of the Japanese government to inhibit marketing such illegal products, the supplement marketplaces outside Japan are still accessible to consumers, who use the Internet as an information source. Thus, Japanese women suffer from adverse events due to the use of those products [5,6].

In the meantime, the use of weight loss supplements accounted for only 2.3% according to the previous research on mothers, which was one tenth of the reports from high school girls. On the other hand, girls who were passive users gained information from their family, and the purpose of use was in line with the report on mothers. The differences between active and passive users suggest that girls who are active users of dietary supplements make their own decisions without their mothers’ involvement. (Line 221-238)

For Paragraph 5, we deleted the repetition and revised as follows: “The prevalence of adverse events in high school ages was shown to be 2.1% according to the report on mothers, which was the lowest occurrence among all the grades of school-aged children, and contradicted the prevalence of dietary supplement use that increased along with age [4]. However, the prevalence of adverse events in this study accounted for 5- to 7-fold of mothers’ reports. Since less than a fourth of them consulted with a family member after the onset of adverse events, they might not have reported a change in their physical condition especially when it was small enough to handle by themselves. Indeed, boys primarily experienced gastrointestinal symptoms that were consistent with the report on mothers, and might not be that serious. However, girls experienced other symptoms that may possibly be a sign of progressing to serious injury like liver failure. In this case, the first step should be to stop using supplements to prevent serious injury. (Line 251-260)

 9. While it is interesting to compare the findings to other studies, it would be important to relate the current findings and their implications on practice/ research in Japan

 (Response) We added “Moreover, in adults, weight-loss supplements are popular among women and performance related supplements are popular among men. These products are often seen to be associated with adverse events. According to the mothers’ understanding of their children, the use of these products among high school students was few. However, by asking high school students directly, it was revealed that even high school students had been in the similar situation with adults.” (Line 271-276)

 Strengths of the paper

10. This is missing in the current manuscript and it is highly recommended to include the strengths besides stating the limitations

 (Response) Thank you for advice. We added the strength after limitation as follows: “However, we conducted the survey covered nationwide on both mothers and children themselves; thus, the difference between the mothers’ understanding and children’s attitudes toward dietary supplements could be defined; surveying on children enabled us to figure out the current circumstances on the use of certain products and adverse events. The findings will provide basic insight to create more suited education programs to mothers and to children, respectively.” (Line 291-296)

 Grammar and spelling

11. Please ensure the manuscript is proof read and edited as there are a number of grammatical errors throughout especially on the tenses used when citing the findings from other studies and selection of words could be improved for the purpose of scientific writing

 (Response) Thank you for suggestion. To undergo MDPI English editing, we need to obtain the permission for the payment from our institute that will take about 2 weeks. At this time, we resubmitted our manuscript without English editing. So, if you suggest it after second review, we will do it as soon as we get a permission.

 Again, we thank the reviewers for giving us the opportunity to revise this manuscript and trust that we have been able to do so to their satisfaction.

 Best regards,

 Tsuyoshi Chiba

Reviewer 2 Report

Thank you for the opportunity to review this well written paper addressing an important topic.

I have the following comments to the manuscript:

Title:

Lines 2-3: Title, I think you should avoid using the terms «active users» and «passive users» without explaining what these terms mean (I did not understand the meaning of these terms until I read the explanation in the manuscript).

Abstract:

Lines 12-13: I think you should avoid using the terms «active users» and «passive users» without explaining what these terms mean (as for the title).

Lines 13-15: When I first read the abstract I thought 46019 persons were invited to the study of whom 1031 answered resulting in a response rate of 2%. Please provide a more precise explanation of how you only aimed to include 1000 students (or rewrite the text to avoid confusion regarding the response rate).

Introduction:

Lines 33-34: I would appreciate knowing the direction of the association between use of supplements in children and the mentioned factors (mother’s age for instance, were older mothers more likely to give their children supplements? If not stated, it could also be that younger mothers were more likely to give their children supplements).

Lines 46-47: I am not familiar with the value of a yen, could you please include some indicator of their comparative value in euros or dollars in addition to the amount in yen?

Materials and methods:

Lines 57-58: Here it says that “The detailed survey procedure has been described previously [13].” As reference number 13 is an article in Japanese, this information is unavailable to many readers. Hence, I would appreciate it if the authors took out this sentence and instead included the necessary information from reference 13 in the present manuscript (for instance how the 46019 invitees were picked out to participate).

Lines 70-71: I didn’t quite understand the phrase “Demographic information such as school grade, resident area, and sports or exercise habits (yes/no) were obtained in the preliminary survey”. Did you have this background information on all 46019 invitees from a previous survey?

Lines 72-73: Why did you decide to group those who reported to previously have used supplements as users? Do you know when they stopped using supplements? Could it be that some of them used supplements as children and therefore reported previous use? It seems a bit misleading to call them “supplement users” when they report to not use supplements when answering the questionnaire. How many of the participants grouped as supplement users reported to be previous users?

Lines 96-97 “demographic information “sports or exercise habits”: Here I think you need to specify that the 38,9% refers to either yes or no to the question about whether or not they were involved in sports or other exercise. Also, I think this sentence is a little difficult to understand: “More supplement users were girls in their 3rd grade (32.5%, p = 0.01), and boys with the demographic information “sports or exercise habits” (38.9%, p = 0.03).” It sounds a bit like girls are more likely to use supplements than boys? Should I interpret it as “among girls, supplement use was more common in the 3rd grade than in the 1st or 2nd grade”? And that “among boys, supplement use was more common among those reporting to be involved in sports or other exercise compared to those reporting not to be involved in sports or other exercise”?

Lines 76 and 106-107: I am no supporter of extended supplement use, but isn’t the general practice of defining “strongly disagree” as the preferred answer a bit too strict? For instance the alternative “Pregnant women should take dietary supplements to supplement nutrition”. In Norway where I am from, all pregnant women are recommended a dietary supplement of folic acid in the first 12 weeks of pregnancy (and when planning pregnancy) to help prevent neural tube defects. Iodine supplements are also recommended if intakes of dairy products and white fish are low. Likewise, vitamin D supplements are recommended if the intake of oily fish is lower than 2-3 times per week. These recommendations are probably a bit different in different countries, but a quick online search showed that folic acid is also recommended in the US, the UK and also in Japan.

Regarding the statement “Dietary supplements can be used concomitantly with drugs”, I find this difficult to answer. There are many supplements and drugs that can be combined without any harm. A general caution when combining supplements and drugs is of course important, but I cannot agree to that all combinations of supplement use and medication use are harmful.

The last statement I would like to comment is the statement “Dietary supplements can treat diseases”. I agree that most diseases cannot be treated by dietary supplements, but diseases caused by deficiencies of one or more nutrients can be treated by supplements. Iron deficiency anemia may for instance be treated using iron supplements. In some diseases, the patient may have malabsorption issues requiring high doses of vitamins and/or minerals.  Supplements are for instance very important after bariatric surgery.

Lines 208-209: I don’t think it is bad to find information on the Internet as long as you pay attention to the source of the information (who has provided the information). Governmental bodies are for instance (hopefully) regarded as trustworthy sources. Manufacturers’ information on the other hand is likely to be less trustworthy.

Lines 239-247: Here I think you need to mention that the number of participants reporting to have experienced adverse events was very low (only 8 boys if I understood correctly?).

Line 268: A minor detail, but isn’t it better to say “parents” than “mothers”? Is it unlikely that fathers may play a role in their children’s supplement use?

Under limitations (lines 261-270): The methods section did not clearly state if the participants knew that they were answering a questionnaire about dietary supplements when they agreed to participate? If they did, the sample might be more interested in supplements than what would be the case in the general population. This may still be an issue even if they did not initially know (as they soon found out when answering the questionnaire). A person interested in supplements might be more willing to complete the questionnaire than a person who is completely uninterested in supplements or diet. I think this should be commented upon in the manuscript.

Also the total number of participating boys (n=276) is very low to say much about the prevalence of supplement use among high school students in a large country like Japan. I think this point deserves a bit more mentioning in the manuscript.

Do you know anything about the educational background or socioeconomic status of their parents? How do they compare to the general population?

Author Response

Answer to Reviewer.2

Thank you for valuable comments and useful suggestion to improve our manuscript. We carefully read them and changed our manuscript by following each comment.

1. Title, I think you should avoid using the terms «active users» and «passive users» without explaining what these terms mean (I did not understand the meaning of these terms until I read the explanation in the manuscript).

 (Response) Thank you for comment. We deleted “active and passive” from the title. In addition, we revised the title to “A nationwide survey on the attitude towards use of dietary supplements amongst Japanese high school students” according to another reviewer’s comment. (Lines 2-3)

 2. Lines 12-13: I think you should avoid using the terms «active users» and «passive users» without explaining what these terms mean (as for the title).

 (Response) We removed “differences between active and passive users” according to the reviewer’s suggestion and added “The participants were classified according to the purchasers of supplements: “oneself” was defined as active users; and others as passive users.” to explain the terms. (Lines 16-17)

 3. Lines 13-15: When I first read the abstract I thought 46019 persons were invited to the study of whom 1031 answered resulting in a response rate of 2%. Please provide a more precise explanation of how you only aimed to include 1000 students (or rewrite the text to avoid confusion regarding the response rate).

 (Response) We requested the research company to collect at least 1000 students. Then, the company decided how many invitation e-mail to send its registrants (in this study, it was 46019). The responses were collected in the first-come basis until the number reached to 1000; thus, the collected number of 1031 was not the total responses from 46019.

Regarding sample size, we planned to collect 600 dietary supplement users of high school students as the same number as our previous survey conducted to mothers of high school students. Assuming the prevalence of supplement users approximately 30% from the previous reports, we requested for quotation for 2000 complete responses from high school students to several research companies which met certain standards. However, a response rate in this age is very low compared to that in general. Indeed, 46,019 registrants were needed to collect 1000 responses in this survey. Therefore, 1000 subjects were the maximum number within the limits of our budget.

To avoid confusion, we removed “46,019” from abstract because we could not explain these details.

 4. Lines 33-34: I would appreciate knowing the direction of the association between use of supplements in children and the mentioned factors (mother’s age for instance, were older mothers more likely to give their children supplements? If not stated, it could also be that younger mothers were more likely to give their children supplements).

 (Response) We revised the sentence to include the direction of the association. “The supplement use in children is shown to be associated with older age of mothers and high levels of education or eating awareness.” (Line 37)

 5. Lines 46-47: I am not familiar with the value of a yen, could you please include some indicator of their comparative value in euros or dollars in addition to the amount in yen?

 (Response) We added the equivalent value in US dollars. “(equivalent to a couple USD to several dozen USD)” after the value of yen. (Line 51)

6. Lines 57-58: Here it says that “The detailed survey procedure has been described previously [13].” As reference number 13 is an article in Japanese, this information is unavailable to many readers. Hence, I would appreciate it if the authors took out this sentence and instead included the necessary information from reference 13 in the present manuscript (for instance how the 46019 invitees were picked out to participate).

 (Response) Thank you for comment. We removed the sentence and revised the survey method to:
“The research company holds over 4 million registrants who are confirmed sociodemographic information annually in Japan and eliminates fraudulent respondents when identified.

We submitted the questionnaire with a request of at least 1,000 responses. The research company sent an invitation e-mail to participate in the survey to its 46,019 registrants (boys: 17,583, girls: 28,436) comprising high school students, aged 15 to 18 years; the number of invitations was decided by the research company based on their predicted response rate. The survey was administered in the following order; the questions started from the general participant questions; then, the further questions were asked only to dietary supplement users” (Lines 64-72)

 7. Lines 70-71: I didn’t quite understand the phrase “Demographic information such as school grade, resident area, and sports or exercise habits (yes/no) were obtained in the preliminary survey”. Did you have this background information on all 46019 invitees from a previous survey?

 (Response) Previous survey was conducted to mothers of high school-aged children. Present survey was conducted independently from the previous one. Thus, the participants were not necessarily the children of mothers from previous study. Invitees accounted for 46019; however, we collected only 1031 responses which included about demographic information.

As it was confusing when we use the word “preliminary survey”, we removed “the preliminary survey” and moved the sentence to the next paragraph describing about the general participant questions. “Demographic information such as school grade, resident area, and sports or exercise habits (yes/no) were also obtained in this general participant questions. (Lines 90-92)

 8. Lines 72-73: Why did you decide to group those who reported to previously have used supplements as users? Do you know when they stopped using supplements? Could it be that some of them used supplements as children and therefore reported previous use? It seems a bit misleading to call them “supplement users” when they report to not use supplements when answering the questionnaire. How many of the participants grouped as supplement users reported to be previous users?

 (Response) We did not ask when the participants stopped using supplements; thus, as the reviewer pointed out, supplement use of previous users could be when they were small children. However, the purpose of this study was to clarify the perceptions and the use of dietary supplements among those who voluntarily use supplements; it was unlikely that small children purchased supplements on their own. In addition, one of our purposes was to know about the occurrence of adverse events; in most cases, people stop using supplements after they experienced adverse events. Therefore, we included previous supplement users as “supplement users”.

We added the numbers of current users, previous users, and non-users: “The general participant questions covered dietary supplement use (the ones who answered “currently using” (n = 127) and “previously used” (n = 159) were considered to be “supplement users,” and “never used” (n = 745) were considered to be “supplement non-users”) and perceptions regarding dietary supplements” (Line 82-83)

 9. Lines 96-97 “demographic information “sports or exercise habits”: Here I think you need to specify that the 38,9% refers to either yes or no to the question about whether or not they were involved in sports or other exercise. Also, I think this sentence is a little difficult to understand: “More supplement users were girls in their 3rd grade (32.5%, p = 0.01), and boys with the demographic information “sports or exercise habits” (38.9%, p = 0.03).” It sounds a bit like girls are more likely to use supplements than boys? Should I interpret it as “among girls, supplement use was more common in the 3rd grade than in the 1st or 2nd grade”? And that “among boys, supplement use was more common among those reporting to be involved in sports or other exercise compared to those reporting not to be involved in sports or other exercise”?

 (Response) As the reviewer commented, the 38.9% in question was the percentage of supplement users who involved in sports or exercise habits in boys (18.2% + 20.7%). Therefore, we changed the sentence to “Among girls, supplement use was seen more in their 3rd grade (32.5%) than in their 1st (24.0%) or 2nd (22.6%) grade (p = 0.01). Among boys, supplement use was seen more among those reporting to be involved in sports or exercise habits (38.9%) compared to those reporting not to be involved in sports or exercise (24.4%) (p = 0.03).” (Line 111-114)

 10. Lines 76 and 106-107: I am no supporter of extended supplement use, but isn’t the general practice of defining “strongly disagree” as the preferred answer a bit too strict? For instance the alternative “Pregnant women should take dietary supplements to supplement nutrition”. In Norway where I am from, all pregnant women are recommended a dietary supplement of folic acid in the first 12 weeks of pregnancy (and when planning pregnancy) to help prevent neural tube defects. Iodine supplements are also recommended if intakes of dairy products and white fish are low. Likewise, vitamin D supplements are recommended if the intake of oily fish is lower than 2-3 times per week. These recommendations are probably a bit different in different countries, but a quick online search showed that folic acid is also recommended in the US, the UK and also in Japan.

 The last statement I would like to comment is the statement “Dietary supplements can treat diseases”. I agree that most diseases cannot be treated by dietary supplements, but diseases caused by deficiencies of one or more nutrients can be treated by supplements. Iron deficiency anemia may for instance be treated using iron supplements. In some diseases, the patient may have malabsorption issues requiring high doses of vitamins and/or minerals.  Supplements are for instance very important after bariatric surgery.

 (Response) Thank you for comment and introducing the situation in Norway. As the reviewer said, the situations and recommendations regarding dietary intake seem to be different depending on countries. In Japan, folic acid is also recommended to pre-pregnant women, but our recommendation comes to take folate from natural food first. If they cannot consume enough amount, then supplement use is encouraged. In addition, most folic acid supplement products contain not only folic acid, but also several ingredients, eg. herbs which do not confirm the safety in mothers and infants. In this situation, we generally recommend regular meals to take folate. Besides, vitamin D and Iodine are generally consumed adequately from regular meals in Japanese pregnant women.

 As the reviewer pointed out, iron deficiency anemia is an important issue in Japan, especially among young women. In Japan, dietary supplements (considered to be foods) and medical drugs are strictly separated. For diseases caused by nutrient deficiencies, patients are prescribed with vitamin and mineral drugs. None of dietary supplements are produced for patients in Japan. But in the actual situation, people do not understand the differences between supplements and drugs, and they use supplements to treat diseases without consulting with health-care professionals.

 To clarify why “strongly disagree” is the most preferable for Japanese, we added a brief explanation in methods as follows: “The most preferred answer was set with the consideration of the current situation in Japan; for example, consuming adequate amounts of nutrients from our regular meals without using health foods including dietary supplements is highly recommended. In addition, dietary supplements and drugs are strictly separated by our regulation; to treat diseases caused by nutrient deficiency, vitamin and mineral drugs are used instead of supplements.” (Lines 85-90)

Again, the situation of food and nutrients intake are quite different among countries. We traditionally eat fish and seaweed which are the major sources of vitamin D and iodine. We also traditionally eat a lot of green leafy vegetable, the source of folate, and our traditional diet has been recognized as one of the healthy diets. However, there is a trend that recent changes in our dietary patterns put away those traditional Japanese diet from our regular meals especially among younger ages. Therefore, as the situation in Norway, we may need to recommend to take supplements in the near future.

 Regarding the statement “Dietary supplements can be used concomitantly with drugs”, I find this difficult to answer. There are many supplements and drugs that can be combined without any harm. A general caution when combining supplements and drugs is of course important, but I cannot agree to that all combinations of supplement use and medication use are harmful.

 (Response) As the reviewer point out, there are certain combinations of nutritional supplements and drugs that bring no harm. If a patient needs additional nutrients under medication, the nutritional supplements will be helpful. Our former researches showed patients tend to overwhelm the safety of supplements. Indeed, a research showed approximately 17.7% to 36.8% of patients in Japan used multiple supplements concomitantly with multiple drugs without telling to their physicians, and another research showed the types of supplements patients used were not only nutritional supplements (41%) but also other herbal, animal-derived or combination supplements (59%) that potential interactions with drugs are not known. Considering the risks of supplement-drug interactions, we recommend not to use supplements concomitantly with drugs.

 11. Lines 208-209: I don’t think it is bad to find information on the Internet as long as you pay attention to the source of the information (who has provided the information). Governmental bodies are for instance (hopefully) regarded as trustworthy sources. Manufacturers’ information on the other hand is likely to be less trustworthy.

 (Response) Referring the source of information is important when obtaining information from the Internet. Information posted on the websites managed by governmental agencies is certainly trustworthy. But unfortunately, few consumers visit those websites to seek information about dietary supplements. Instead, most consumers visit websites that provide attractive information, such as supplement manufacturers’ and retailers’ websites, blogs and personal websites of popular artists, and online community to share ideas and experiences. Information on these websites is typically not provided based on scientific evidence.

However, we did not ask which website they referred for supplement information, we deleted the sentence: “They gained information through the Internet, which is not always the right medium for such information” to avoid mislead.

 12. Lines 239-247: Here I think you need to mention that the number of participants reporting to have experienced adverse events was very low (only 8 boys if I understood correctly?).

 (Response) The reviewer’s recognition is right. Only 8 boys experienced adverse events. We also think the number (n=8) is quite low, so this result cannot be interpreted as general population. Though it was lower than girls, the prevalence was 9.4% (8/85) and this percentage is consistent with the previous researches in general. In addition, the survey on mothers could not gain even in this prevalence. Therefore, we could not ignore this findings although these should be confirmed in further studies.

Thus, we added a description as a limitation: “In addition, adverse event information from the small number of boys could not be interpreted as general population of supplement users.” (Line 290-291)

 13. Line 268: A minor detail, but isn’t it better to say “parents” than “mothers”? Is it unlikely that fathers may play a role in their children’s supplement use?

 (Response) Since our previous survey was conducted only to mothers, we stuck to discussing mothers; however, in the present survey, we asked who the purchaser of supplements is with the answering options of “oneself”, “parents,” “grandparents,” and “distributed in the school program”. Because we cannot determine who in the passive users. We changed the sentence as follows: “Second, the information on the unintentional supplement use by participants may not have been included, especially for passive users who did not pay attention to what they were given.” (Line 287)

 14. Under limitations (lines 261-270): The methods section did not clearly state if the participants knew that they were answering a questionnaire about dietary supplements when they agreed to participate? If they did, the sample might be more interested in supplements than what would be the case in the general population. This may still be an issue even if they did not initially know (as they soon found out when answering the questionnaire). A person interested in supplements might be more willing to complete the questionnaire than a person who is completely uninterested in supplements or diet. I think this should be commented upon in the manuscript.

 (Response) We agree with the reviewer’s comment that a person interested in supplements could be more willing to complete the questionnaire. Indeed, invitation e-mail contained the survey theme about dietary supplement. However, the main purpose of this study was to clarify the attitudes toward dietary supplement in this age, but not the prevalence of supplement use. In this regard, we think that willingness of participating this survey is not critical limitation in this survey.

 15. Also the total number of participating boys (n=276) is very low to say much about the prevalence of supplement use among high school students in a large country like Japan. I think this point deserves a bit more mentioning in the manuscript.

Do you know anything about the educational background or socioeconomic status of their parents? How do they compare to the general population?

 (Response) As the reviewer pointed out, We added “However, the response rate was very low in this age group; the participants were not necessarily representatives of Japanese high school students. Indeed, the research company estimated the almost 40,000 samples to get 1,000 actual responders.” (Line 281-284)

 Again, we thank the reviewers for giving us the opportunity to revise this manuscript and trust that we have been able to do so to their satisfaction.

 Best regards, 

Tsuyoshi Chiba

Round  2

Reviewer 2 Report

Dear authors,

Thank you for addressing my comments, now I think it is much easier to understand how you conducted the study and some of the differences between Japan and the countries I am more familiar with. To further enhance the quality of the manuscript I would recommend that you show it to a native English speaking person to make sure all your sentences are properly formulated.

Author Response

Answer to Reviewer.2

 Thank you for reviewing our manuscript and suggesting English editing.

The revised manuscript has been undergone English editing by MDPI.

 Again, we thank the reviewer for giving us the opportunity to improve this manuscript.

 Best regards,

Tsuyoshi Chiba
